# Exponential Hardness of Optimization from the Locality in Quantum Neural Networks

## Abstract

Quantum neural networks (QNNs) have become a leading paradigm for establishing near-term quantum applications in recent years. The trainability issue of QNNs has garnered extensive attention, spurring demand for a comprehensive analysis of QNNs in order to identify viable solutions. In this work, we propose a perspective that characterizes the trainability of QNNs based on their locality. We prove that the entire variation range of the loss function via adjusting any local quantum gate vanishes exponentially in the number of qubits with a high probability for a broad class of QNNs. This result reveals extra harsh constraints independent of gradients and unifies the restrictions on gradient-based and gradient-free optimizations naturally. We showcase the validity of our results with numerical simulations of representative models and examples. Our findings, as a fundamental property of random quantum circuits, deepen the understanding of the role of locality in QNNs and serve as a guideline for assessing the effectiveness of diverse training strategies for quantum neural networks.

## 1 Introduction

Quantum computing is a rapidly growing technology that exploits quantum mechanics to solve intricate problems that classical computers cannot solve. With enormous efforts having been made to develop noisy intermediate scale quantum (NISQ) devices [1], current quantum devices have demonstrated the ability to achieve near-term quantum advantage for practical applications in key areas including many-body physics [2–4], chemistry [5], finance [6–8], and machine learning [9]. Specifically, quantum machine learning (QML) represents an exciting, emerging interdisciplinary field that seeks to enhance machine learning algorithms by harnessing the inherent parallelism of quantum systems [10–20]. Quantum neural networks (QNNs) stand at the forefront of QML, capitalizing on the unprecedented potential of quantum computing to revolutionize data analysis and pattern recognition. Inspired by classical neural networks, QNNs employ quantum gates and quantum states as fundamental building blocks within their computational framework. These networks can be trained using a diverse range of methods, including gradient-based optimization techniques akin to classical neural network training [21–24].

With the aim to show quantum advantage on certain tasks, a critical issue is whether QNNs can be extended to solve large-scale systems, i.e., scalability. Unfortunately, many studies point out that training of QNNs requires exponential resources with the system size under certain conditions [25–36]. Besides the practical limitations such as noises [29], even ideal quantum devices will suffer from the so-called *barren plateau* phenomenon [25], which is the quantum counterpart of vanishing gradient problem in classical machine learning. It was shown that the gradient of the cost function vanishes exponentially in the number of qubits with a high probability for a random initialized QNN with sufficient depth, analogous to the vanishing gradient issue in classical neural networks.

Submitted to 37th Conference on Neural Information Processing Systems (NeurIPS 2023). Do not distribute.

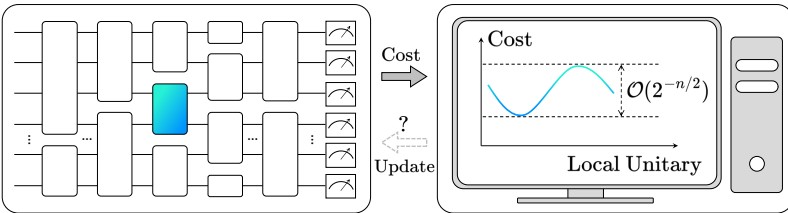

Figure 1: **Training limitations from QNN locality.** The left part depicts a PQC on $n$ qubits composed of local unitaries. The right part symbolically depicts the cost function on a classical device vs. the local unitary highlighted in the left part. This work proves that the cost function will fluctuate in an exponentially small range in the number of qubits with a high probability when we vary an arbitrary local unitary within the QNN in certain cases.

Consequently, exponentially vanishing gradients demand exponential precision in the cost function measurement on a quantum device [37] to make progress in the gradient-based optimization, and hence an exponential complexity in the number of qubits.

Several attempts have been made to avoid barren plateaus, such as higher order derivatives [38], gradient-free optimizers including gate-by-gate optimization [39, 40], proper initialization [41], pre-training including adaptive methods [42–46], QNN architectures [47, 48] and cost function choices [49, 50]. More efforts are needed to study the general effectiveness of these attempts [26, 27] and develop new strategies to improve the trainability and scalability of QNNs. As a guide for exploring effective training strategies, it is crucial to uncover the essential mechanisms behind the barren plateau phenomenon.

However, few rigorous scaling results are known for generic QNNs besides phenomenological calculations, i.e., gradient analyses and their descendent [26–28]. Instead of just the limited information of vicinity from gradient analyses, it would be quite helpful for designing efficient algorithms if we could gain information on the entire variation range of the cost function when adjusting a single [39, 40] or several parameters. Combined with the fact that parameters usually enter the circuit independently through local quantum gates, all of which motivate our work where we are chiefly concerned with the variation range of the cost function via varying a local unitary within a quantum circuit.

In this work, we present a rigorous scaling theorem on the trainability of QNNs beyond gradients from the perspective of QNN locality. As summarized in Fig. 1, we prove that when varying a *local unitary* within a sufficiently random circuit, the expectation and variance of the variation range of the cost function vanish exponentially in the number of qubits. Then through simple derivations, we show that this theorem implies exponentially vanishing gradients and cost function differences, and hence unifies the restrictions on gradient-based and gradient-free optimizations. Meanwhile, this theorem further delivers extra meaningful information about the training landscapes and optimization possibilities of QNNs. In this sense, we obtain a fundamental limitation on QNN training. Next, we illustrate the applications of our theorem on representative QNN models, where a tighter bound for the fidelity-type cost function is provided specifically even with shallow random circuits. At last, we perform numerical simulations on these representative models, where the scaling exponents coincide with our analytical results almost precisely.

**Comparison with Previous Works.** The advances of our results compared to previous works [25, 27, 26, 28] exist in two aspects. Firstly, the exponentially vanishing quantity we claim is the *entire* variation range of the cost function in the *whole parameter subspace* corresponding to the local unitary. This provides constraints on multiple parameters at finite intervals simultaneously, instead of an infinitesimal vicinity or two fixed-parameter points. Secondly, our results are irrelevant with the parameterization of the local unitary like $e^{-i\Omega\theta}$ used previously. Hence, our results are much more general whose only condition is the circuit locality and open a new avenue for analyzing the QNN trainability.

## 2 Preliminaries

**Quantum State.** We first introduce basic concepts and notations in quantum computing. A pure single-qubit quantum state is a linear combination of two computational basis states, represented as $|\phi\rangle = \alpha|0\rangle + \beta|1\rangle$ in Dirac notation, where $\alpha, \beta \in \mathbb{C}, |\alpha|^2 + |\beta|^2 = 1$. Here, $|0\rangle$ and $|1\rangle$ denote the basis states $[1, 0]^T$ and $[0, 1]^T$ in the single-qubit space $\mathbb{C}^2$, respectively. The $n$-qubit space $\mathbb{C}^{2^n}$ is formed by the tensor product of $n$ single-qubit spaces. Additionally, the quantum state can be represented by a positive semidefinite matrix, also known as a density matrix. The density matrix $\rho$ of a pure state $|\phi\rangle$ consisting of $n$ qubits is expressed as $\rho = |\phi\rangle\langle\phi|$, where $\langle\phi| = |\phi\rangle^\dagger$. A general mixed quantum state is represented by $\rho = \sum_k c_k|\phi_k\rangle\langle\phi_k|$, where $c_k \in \mathbb{R}, \sum_k c_k = 1$.

**Quantum Gate.** Quantum gates are mathematically described as unitary operators. Common single-qubit gates include the Pauli rotations $\{R_P(\theta) = e^{-i\frac{\theta}{2}P}|P \in \{X, Y, Z\}\}$, which are in the matrix exponential form of Pauli matrices

$$X = \begin{pmatrix} 0 & 1 \\ 1 & 0 \end{pmatrix}, \qquad Y = \begin{pmatrix} 0 & -i \\ i & 0 \end{pmatrix}, \qquad Z = \begin{pmatrix} 1 & 0 \\ 0 & -1 \end{pmatrix}. \tag{1}$$

Common two-qubit gates include controlled-X gate $\text{CNOT} = I \oplus X$ ($\oplus$ is the direct sum) and controlled-Z gate $\text{CZ} = I \oplus Z$, which can generate quantum entanglement among qubits.

**Quantum Measurement.** Quantum measurement is a quantum operation to obtain information from the quantum system. For example, for a single-qubit state $|\phi\rangle = \alpha|0\rangle + \beta|1\rangle$, the outcome of a computational basis measurement is either $|0\rangle$ with probability $|\alpha|^2$ or $|1\rangle$ with probability $|\beta|^2$. This measurement operation can be mathematically referred to as the average of the observable $O = Z$ under the state $|\phi\rangle$: $\langle\phi|O|\phi\rangle = \text{tr}[Z|\phi\rangle\langle\phi|] = |\alpha|^2 - |\beta|^2$. Generally, quantum observables $O$ are Hermitian matrices and $\mathcal{O}(1/\varepsilon^2)$ times of measurements could give an $\varepsilon\|O\|_\infty$-error estimation to the value $\text{tr}[O\rho]$, where $\|\cdot\|_\infty$ is the spectral norm of the matrix.

**Quantum Neural Network.** While classical neural networks operate on classical bits and use classical logic gates, quantum neural networks (QNNs) use quantum bits, or qubits, and quantum gates to process and store information. QNNs are often described as parameterized quantum circuits (PQCs) that are composed of rotation gates with adjustable rotating angles. In general, a QNN takes the mathematical form $\mathbf{U}(\boldsymbol{\theta}) = \prod_\mu U_\mu(\theta_\mu)W_\mu$, where $U_\mu(\theta_\mu) = e^{-i\theta_\mu\Omega_\mu}$ denotes a parameterized gate, such as a single-qubit rotation gate with $\Omega_\mu$ representing a Hermitian operator, and $W_\mu$ corresponds to fixed gates like the CNOT gate and SWAP gate. Commonly used templates of QNNs include the hardware efficient ansatz, the alternating-layered ansatz, and the tensor-network-based ansatz [49, 51]. Note that QNNs with intermediate classical controls such as QCNNs [52] can also be included in this general form theoretically.

## 3 Limitations of Local Unitary Optimization in QNN

We start by introducing a general setting of a QNN model used throughout our analysis. A hybrid quantum-classical framework in QML usually uses a classical optimizer to train a QNN, denoted by $\mathbf{U}$, with an input state $\rho$ by minimizing a task-dependent cost function $C$, which is typically chosen as the expectation value of some Hermitian operator $H$:

$$C_{H,\rho}(\mathbf{U}) = \text{tr}(H\mathbf{U}\rho\mathbf{U}^\dagger). \tag{2}$$

Note that other cost function forms can be regarded as compositions of observable expectations and some other classical post-processing functions. Here we focus on (2) for simplicity. Divide the whole qubit system into two parts $A, B$ with $m$ qubits and $n - m$ qubits, respectively. Here $m$ is a fixed constant not scaling with $n$ so that we call $A$ a local subsystem. The QNN $\mathbf{U}$ is often composed of local unitaries on real devices, such as the single-qubit rotation gates and the CNOT gate. We focus on a local unitary $U_A$ within $\mathbf{U}$ acting on subsystem $A$. As shown in Fig. 2, we denote the sub-circuit of $\mathbf{U}$ before $U_A$ as $V_1$ and that behind $U_A$ as $V_2$, such that $\mathbf{U} = V_2(U_A \otimes I_B)V_1$ where $I_B$ is the identity operator on $B$. $V_1, V_2$ and $U_A$ are independent of each other. We also remark that this circuit setting is sufficiently general to cover common representative QNN models, e.g., the variational quantum eigensolver, the quantum autoencoder, and the quantum state learning.

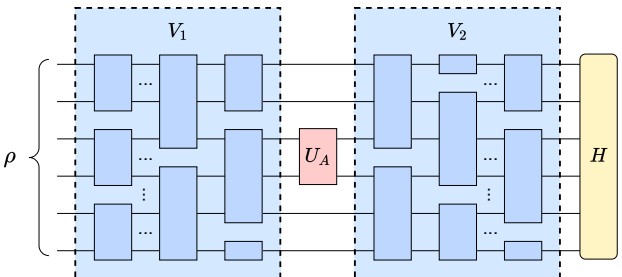

Figure 2: **Partition of the QNN in our analysis**. The QNN is decomposed as $\mathbf{U} = V_2(U_A \otimes I_B)V_1$ with an input state $\rho$ and an observable $H$. A tunable local unitary $U_A$ is implemented by some local quantum gates with the left and right parts assembled as $V_1$ and $V_2$.

To characterize the training landscape beyond the limited information of the vicinity from gradient analyses, we introduce a central quantity throughout this work, i.e., the *variation range of the cost function* via varying a local unitary.

**Definition 1** *For a generic cost function $C_{H,\rho}(\mathbf{U})$ with a QNN $\mathbf{U}$ in Eq. (2), we define its variation range with given $V_1, V_2$ as*

$$\Delta_{H,\rho}(V_1, V_2) := \max_{U_A} C_{H,\rho}(\mathbf{U}) - \min_{U_A} C_{H,\rho}(\mathbf{U}), \tag{3}$$

*where the maximum and minimum with respect to $U_A$ are taken over the unitary group $\mathcal{U}(2^m)$ of degree $2^m$.*

The quantity $\Delta_{H,\rho}(V_1, V_2)$ intuitively reflects the maximal possible influence that the local unitary $U_A$ can have on the cost function. We establish an upper bound on $\Delta_{H,\rho}(V_1, V_2)$ in the sense of probability by Theorem 1, which thus delivers a limitation on optimizing an arbitrary local unitary. To be specific, we prove that if either $V_1$, $V_2$, or both match the Haar distribution up to the second moment, i.e., are sampled from unitary 2-designs [53], the expectation of $\Delta_{H,\rho}(V_1, V_2)$ vanishes exponentially in the number of qubits. See Appendix A for preliminaries on unitary designs.

**Theorem 1** *Suppose $\mathbb{V}_1, \mathbb{V}_2$ are ensembles from which $V_1, V_2$ are sampled, respectively. If either $\mathbb{V}_1$ or $\mathbb{V}_2$, or both form unitary 2-designs, then for arbitrary $H$ and $\rho$, the following inequality holds*

$$\mathbb{E}_{V_1,V_2}[\Delta_{H,\rho}(V_1, V_2)] \leq \frac{w(H)}{2^{n/2-3m-2}}, \tag{4}$$

*where $\mathbb{E}_{V_1,V_2}$ denotes the expectation over $\mathbb{V}_1, \mathbb{V}_2$ independently. $w(H) = \lambda_{\max}(H) - \lambda_{\min}(H)$ denotes the spectral width of $H$, where $\lambda_{\max}(H)$ is the maximum eigenvalue of $H$ and $\lambda_{\min}(H)$ is the minimum.*

Theorem 1 demonstrates that the maximal influence of a local unitary within a random QNN on the cost function diminishes exponentially in the number of qubits, with a high probability. This inherent locality of QNN poses an exponential hardness of optimization in QNN training and we would like to make several remarks to better reveal the underlying implications of the theorem below. The main proof idea of Theorem 1 is to calculate the expectation value over $\mathbb{V}_1, \mathbb{V}_2$ separately. To tackle the maximization over $U_A$, the main technique is to employ Hölder's inequality to extract $U_A$ out and bound the remaining part with specific calculations of 2-design element-wise integrals. For the detailed proof, we defer to Appendix B.

**Remark 1** Firstly, due to the non-negativity and boundedness of the variation range, i.e., $\Delta_{H,\rho} \in [0, w(H)]$, the variance of $\Delta_{H,\rho}$ can be bounded by its expectation times $w(H)$. Thus from Theorem 1 we know that the variance also vanishes exponentially:

$$\mathrm{Var}_{V_1,V_2}[\Delta_{H,\rho}(V_1, V_2)] \leq \frac{w^2(H)}{2^{n/2-3m-2}}. \tag{5}$$

Note that $w(H) \in \mathcal{O}(\text{poly}(n))$ holds for common VQAs. Moreover, Theorem 1 together with Markov's inequality provides an exponentially small upper bound of the probability that $\Delta_{H,\rho}(V_1, V_2)$ deviates from zero, i.e.,

$$\Pr[\Delta_{H,\rho}(V_1, V_2) \geq \epsilon] \leq \frac{1}{\epsilon} \cdot \frac{w(H)}{2^{n/2-3m-2}}, \forall \epsilon > 0. \tag{6}$$

That is to say, the probability that $\Delta_{H,\rho}$ is non-zero to some fixed precision is exponentially small.

**Remark 2** Secondly, we can even establish an exponentially small bound using Theorem 1 for the case where $U_A$ is a *global unitary* satisfying the parameter-shift rule [54–58]. Suppose $U_A = e^{-i\theta\Omega}$ with the Hermitian generator $\Omega$ satisfying $\Omega^2 = I$. Since $\Omega$ has only two different eigenvalues $\pm 1$, there exists a unitary $W$ such that $We^{-i\theta\Omega}W^\dagger$ becomes a local unitary acting on a single qubit non-trivially. $W$ and $W^\dagger$ could be absorbed into the rest of the circuit with $W^\dagger \mathbb{V}_1$ or $\mathbb{V}_2 W$ still forming 2-designs [59]. Therefore, the proof for global unitaries satisfying the parameter-shift rule can be reduced back to the case of local unitaries.

**Remark 3** Moreover, it is worth noticing that the compact bound in (4) only involves the spectral width $w(H)$ and does not depend on any detail of the Hermitian operator $H$. But if some specific structures about $H$ are known, e.g., the Pauli decomposition of $H$, a tighter bound could be derived in Appendix B which depends on the coupling complexity of $H$. In addition, if the cost function reduces to the form of the fidelity between pure states, we could have a tighter bound with scaling $\mathcal{O}(2^{-n})$ in Proposition 2. Theorem 1 can be generalized to arbitrary dimensions besides qubit systems of dimension $2^n$, e.g., qutrit and qudit systems. The detailed proof is provided in Appendix B.

In fact, Theorem 1 has a natural physical interpretation: the effect of a local operation on a physical observable will vanish exponentially after a chaotic evolution. Remarkably, the concept of local operations yielding minor global influences is a physically intuitive yet mathematically intricate notion. For instance, even a single-qubit unitary is enough to rotate an arbitrary $n$-qubit pure state to a new state with zero fidelity with the original one, showcasing local operations do make a great global influence. Hence, Theorem 1 may be invaluable as a rigorous formulation of the aforementioned argument within the domain of QNN training, elucidating the locality of QNNs.

## 4 Unifying the Limitations on Training QNNs

Here we briefly demonstrate how Theorem 1 unifies the restrictions on gradient-based [25, 27] and gradient-free optimizations [26, 28] in a more natural manner, and indicates the extra restrictions besides them on QNN training. In the following, we focus on a PQC applicable for Theorem 1 with $M$ trainable parameters $\{\theta_\mu\}_{\mu=1}^M$ and denote the variation range of the cost function via varying $\theta_\mu$ as $\Delta_\mu$.

Consider the gradient-based optimization first. On the one hand, in the case where the parameter-shift rule is valid [54–58], Theorem 1 can strictly deduce vanishing gradients. Suppose $\{\theta_\mu\}_{\mu=1}^M$ are applicable for the parameter-shift rule (e.g., hardware-efficient ansatzes). Namely, $\theta_\mu$ enters the unitary $e^{-i\theta_\mu\Omega_\mu}$ within the circuit where $\Omega_\mu$ is a Hermitian generator satisfying $\Omega_\mu^2 = I$. From Theorem 1 we know that the expectation of $\Delta_\mu$ vanishes exponentially. Therefore, the derivative $\partial_\mu C := \frac{\partial C}{\partial \theta_\mu}$ with respect to $\theta_\mu$ satisfies

$$\mathbb{E}[|\partial_\mu C|] = \mathbb{E}\left[\left|C\left(\boldsymbol{\theta} + \frac{\pi}{4}\mathbf{e}_\mu\right) - C\left(\boldsymbol{\theta} - \frac{\pi}{4}\mathbf{e}_\mu\right)\right|\right] \leq \mathbb{E}[\Delta_\mu] \in \mathcal{O}(2^{-n/2}), \tag{7}$$

where $\mathbf{e}_\mu$ is the unit vector in the parameter space corresponding to $\theta_\mu$. From Markov's inequality as in (6), we know that the probability that the derivative $\partial_\mu C$ deviates from zero by a small constant is exponentially small.

On the other hand, even in the absence of the parameter-shift rule, vanishing gradients could still be obtained approximately by the following arguments. Consider the vicinity of a random initialized parameter point where the linear approximation error is negligible, denoted as an $\varepsilon$-ball $\mathcal{B}_\varepsilon$ of radius $\varepsilon$ (here $\varepsilon$ plays the same role as the learning rate). As shown in Fig. 3, the linearity in $\mathcal{B}_\varepsilon$ together with Theorem 1 leads to

$$\mathbb{E}[|\partial_\mu C|] \leq \mathbb{E}\left[\frac{\Delta_\mu}{2\varepsilon}\right] \in \mathcal{O}(2^{-n/2}\frac{1}{\varepsilon}), \tag{8}$$

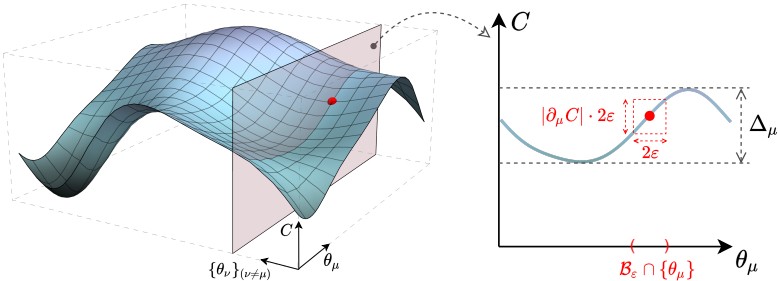

Figure 3: **Sketch of our results implying vanishing gradients.** The left panel sketches the whole training landscape with one of the parameters $\theta_\mu$ as the $x$-axis, all of the other parameters $\{\theta_\nu\}_{\nu\neq\mu}$ as the $y$-axis symbolically and the cost function value $C$ as the $z$-axis. The right panel depicts a typical sample of the $z$-$x$ cross-section from the landscape on the left with variation range $\Delta_\mu$. Up to the linear approximation error, $\Delta_\mu$ serves as an upper bound for the absolute derivative $|\partial_\mu C|$ times the vicinity size $2\varepsilon$.

up to the linear approximation error, where $1/\varepsilon$ is not an essential factor since it reflects the frequencies of the landscape fluctuation rather than magnitudes, similar to the role of the factor $\mathrm{tr}(V^2)$ in the expression of $\mathrm{Var}[\partial_\mu C]$ [25].

For the gradient-free optimization based on the cost function difference between any two *fixed* parameter points $\boldsymbol{\theta}'$ and $\boldsymbol{\theta}$, Theorem 1 leads to

$$\mathbb{E}\left[|C(\boldsymbol{\theta}') - C(\boldsymbol{\theta})|\right] \leq \mathbb{E}\left[\sum_{\mu=1}^{M}\left|C\left(\boldsymbol{\theta}^{(\mu)}\right) - C\left(\boldsymbol{\theta}^{(\mu-1)}\right)\right|\right] \leq \sum_{\mu=1}^{M}\mathbb{E}\left[|\Delta_\mu|\right] \in \mathcal{O}(M2^{-n/2}), \quad (9)$$

where $\boldsymbol{\theta}^{(\mu)} = \boldsymbol{\theta} + \sum_{\nu=1}^{\mu}(\theta'_\nu - \theta_\nu)\,\mathbf{e}_\nu$ for $\mu = 1, ..., M$ and $\boldsymbol{\theta}^{(\mu)} = \boldsymbol{\theta}$ for $\mu = 0$. Thus, as long as the number of parameters satisfies $M \in \mathcal{O}(\mathrm{poly}(n))$, the cost function difference between any two points vanishes exponentially with a high probability, demanding an exponential precision to make progress in the gradient-free optimization.

Furthermore, Theorem 1 goes beyond vanishing gradients and vanishing differences between two fixed points. The exponentially vanishing quantity claimed by Theorem 1 is the variation range of the cost function in the *whole parameter subspace* corresponding to a local unitary, e.g., the subspace of the 3 Euler angles in a single-qubit rotation gate from $\mathcal{SU}(2)$, or the subspace of the 15 parameters in a two-qubit rotation gate from $\mathcal{SU}(4)$, etc. This gives constraints on multiple parameters at finite intervals simultaneously, instead of a vicinity or two fixed parameter points.

## 5 Application on Representative QNN Models

To better illustrate the meaning of our findings in practice, we investigate the applications of Theorem 1 on three representative QNN models, including the variational quantum eigensolver (VQE), quantum autoencoder, and quantum state learning. The corresponding numerical simulation results are summarized in Fig. 5.

**Application on VQE.** The variational quantum eigensolver is the most famous implementation of a hybrid quantum-classical algorithm with the goal to prepare the ground state of a given Hamiltonian $\hat{H}$ of a physical system [60]. The cost function is the energy expectation with respect to an ansatz state $\mathbf{U}|0\rangle$, i.e.,

$$C_{\mathrm{VQE}}(\mathbf{U}) = \langle 0|\mathbf{U}^\dagger \hat{H} \mathbf{U}|0\rangle. \quad (10)$$

For most physical models with local interactions, the spectral width is proportional to the system size, i.e., $w(\hat{H}) \in \mathcal{O}(n)$. For common repeated-layer-type ansatzes, e.g., the hardware-efficient ansatzes [61], linear depth $\mathcal{O}(n)$ is enough to make a randomly initialized circuit to be a sample from an approximate 2-design ensemble [25, 62, 63]. Hence from Theorem 1 we know that $\Delta_{\mathrm{VQE}}(V_1, V_2)$ vanishes exponentially with a high probability for random circuits forming 2-designs. We conduct

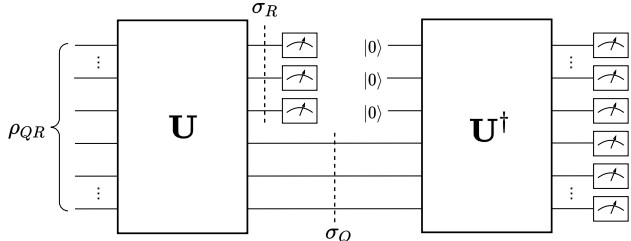

Figure 4: **Circuit setting of the quantum autoencoder.** $\rho_{QR}$ is the given state to be compressed and $\sigma_Q$ is the compressed state through the encoder $\mathbf{U}$. The quantum autoencoder aims to train $\mathbf{U}$ such that $\rho_{QR}$ can be reconstructed from $\sigma_Q$ with high fidelity through the decoder $\mathbf{U}^\dagger$ combined with an ancilla zero state $|0\rangle\langle0|_R$. $\sigma_R$ denotes the state of the discarded part after compression.

numerical simulations for the variation range of the VQE cost function $\Delta_{\mathrm{VQE}}$ using the 1-dimensional spin-$1/2$ antiferromagnetic Heisenberg model:

$$\hat{H} = \sum_{i=1}^{n} \left( X_i X_{i+1} + Y_i Y_{i+1} + Z_i Z_{i+1} \right), \tag{11}$$

with periodic boundary condition, as shown in Fig. 5(a).

**Application on Quantum Autoencoder.** The quantum autoencoder (QAE) is an approach for quantum data compression [64, 65]. As shown in Fig. 4, a QNN $\mathbf{U}$ is trained as an encoder to compress a given state $\rho_{QR}$ on a bipartite system $QR$ into a reduced state $\sigma_Q = \mathrm{tr}_R(\mathbf{U}\rho_{QR}\mathbf{U}^\dagger)$ on subsystem $Q$, such that $\rho_{QR}$ can be reproduced from $\sigma_Q$ by the decoder isometry $\langle0|_R\mathbf{U}^\dagger$ with a high fidelity. According to the monotonicity of the fidelity under partial trace, an easy-to-measure cost function could be reduced from the fidelity between $\rho_{QR}$ and the reconstructed state as

$$C_{\mathrm{QAE}}(\mathbf{U}) := 1 - \mathrm{tr}\left( (|0\rangle\langle0|_R \otimes I_Q) \mathbf{U}\rho_{QR}\mathbf{U}^\dagger \right), \tag{12}$$

where the second term is exactly the fidelity between the state of the discarded part $\sigma_R = \mathrm{tr}_Q(\mathbf{U}\rho_{QR}\mathbf{U}^\dagger)$ and the zero state $|0\rangle_R$ on subsystem $R$. The spectral width for the QAE cost function (12) is $w(H_{\mathrm{QAE}}) = 1$ with $H_{\mathrm{QAE}} = I_{QR} - |0\rangle\langle0|_R \otimes I_Q$. Thus again from Theorem 1 we know that $\Delta_{\mathrm{QAE}}(V_1, V_2)$ vanishes exponentially in the number of qubits, specifically with the scaling $\mathcal{O}(2^{-n/2})$ as shown in Fig. 5(b).

**Application on Quantum State Learning.** The fidelity between pure states is a special case of the cost function in (2) with a low-rank observable. Many QML applications make use of fidelity as their cost functions [66–68]. Here we uniformly call them quantum state learning (QSL) tasks. Denote the input state as $|\psi\rangle$ and the target state as $|\phi\rangle$. The QSL cost function can be written as

$$C_{\mathrm{QSL}}(\mathbf{U}) = 1 - |\langle\phi|\mathbf{U}|\psi\rangle|^2. \tag{13}$$

Theorem 1 can be applied here with $H_{\mathrm{QSL}} = I - |\phi\rangle\langle\phi|$ and $w(H_{\mathrm{QSL}}) = 1$. Here a tighter bound for $\Delta_{\mathrm{QSL}}$ is provided in Proposition 2, which generally holds for the Bures fidelity. The proof of Proposition 2 is detailed in Appendix C.

**Proposition 2** *If either $\mathbb{V}_1$ or $\mathbb{V}_2$, or both form unitary 1-designs, then for the variation range of the fidelity-type cost function $\Delta_{\mathrm{QSL}}$, the following inequality holds*

$$\mathbb{E}_{V_1, V_2} \left[ \Delta_{\mathrm{QSL}}(V_1, V_2) \right] \leq \frac{1}{2^{n-2m}}. \tag{14}$$

Compared with Theorem 1, the bound $\mathcal{O}(2^{-n})$ becomes tighter and the demanded randomness becomes weaker in this special case. Notably, even a random circuit of constant depth is enough to form a 1-design, which is much shallower than 2-designs. Like in (5) and (6), the variance and the probability that $\Delta_{\mathrm{QSL}}$ deviates from zero also vanish exponentially, but only require random circuits forming unitary 1-designs. Moreover, still with 1-designs, Proposition 2 implies exponentially vanishing cost gradients and cost differences in the same way as Theorem 1, which may be considered as the underlying mechanism behind the severe barren plateaus for global cost functions even with shallow quantum circuits [49].

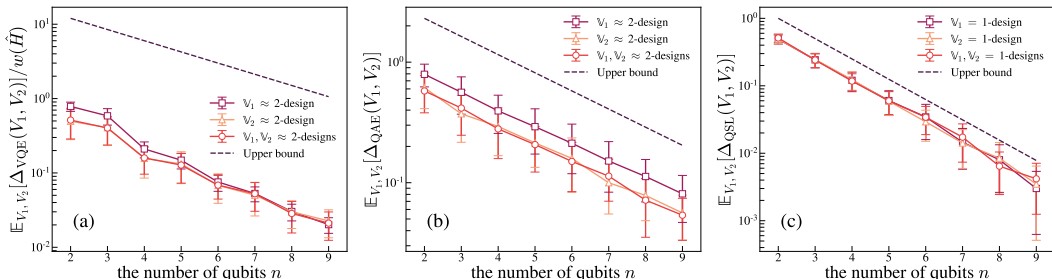

Figure 5: **Exponentially vanishing variation range of the cost function via varying a local unitary.** The data points represent the sample averages of the cost variation range $\Delta_{H,\rho}$ via varying a single-qubit unitary over the spectral width $w(H)$ as a function of the number of qubits on semi-log plots. Panel (a) and (b) correspond to the VQE with the 1-dimensional Heisenberg model and the quantum autoencoder with one qubit discarded, respectively, where the error bars represent the standard deviations over samples. Panel (c) corresponds to the quantum state learning with the cost function being the fidelity with the zero state. Different legends stand for $\mathbb{V}_1$, $\mathbb{V}_2$ or both being approximate 2-designs in (a), (b) and 1-designs in (c). The dashed lines depict our theoretical upper bounds for the three tasks where the scaling exponents show a good coincidence with the experimental results.

## 6 Numerical Simulations of Experiments

Previously, we have theoretically shown that with a high probability, the maximal influence of a local unitary within a random QNN on the cost function will vanish exponentially in the number of qubits. We further demonstrate the validity of our results with numerical simulations of experiments on the three representative QNN models. All of these experimental results show the exponentially vanishing variation range in the number of qubits, which is consistent with Theorem 1 and Proposition 2.

**Circuit Setting.** Consider subsystem $A$ only containing a single qubit, namely $m = 1$, and parameterize the local unitary $U_A \in \mathcal{U}(2)$ with 3 Euler angles up to a global phase, i.e., $U_A(\phi, \theta, \alpha) = R_z(\phi)R_y(\theta)R_z(\alpha)$, where $R_y$ and $R_z$ are single-qubit rotation gates with generators being $Y$ and $Z$ Pauli matrices. To construct random circuits forming 2-designs as $V_1$ or $V_2$ used in the VQE and QAE examples, we employ the following hardware-efficient ansatz as in [25] for comparison.

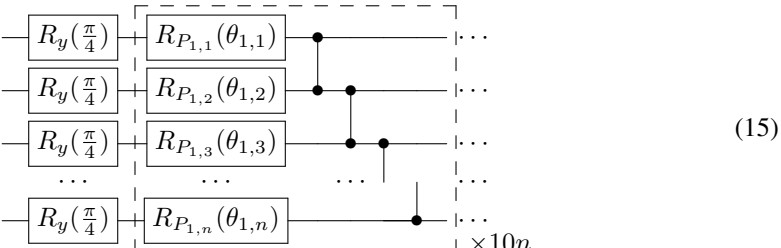

(15)

A single layer of $R_y(\pi/4) = \exp(-iY\pi/8)$ gates are laid at the very beginning of the circuit to make the three rotation axes have equal status, then followed by $10 \times n$ repeated layers. Each layer consists of $n$ single-qubit rotation gates $R_P(\theta)$ on each qubit together with $n - 1$ controlled phase gates between nearest neighboring qubits aligned as a 1-dimensional array, where the rotation axes $P \in \{x, y, z\}$ is chosen with uniform probability and $\theta \in [0, 2\pi)$ is also chosen uniformly. A such random circuit with $\mathcal{O}(n)$ repeated layers could be considered as an approximate 2-design (here we employ $10 \times n$) [25, 62, 63]. Experimental results with different numbers of layers are also presented in Appendix D to show how the expectation of the cost variation range $\Delta_{H,\rho}$ vanishes with the circuit depth. To construct random circuits forming 1-designs used in the QSL example, we just replace the repeated layers above with a single layer of $\mathcal{SU}(2)$ elements $R_z(\phi)R_y(\theta)R_z(\alpha)$ on each qubit with $\phi, \theta, \alpha \in [0, 2\pi)$ are chosen with uniform probability.

**Implementation Details.** To compute $\max_{U_A} C$ and $\min_{U_A} C$ in the definition of $\Delta_{H,\rho}(V_1, V_2)$ with respect to $U_A$, we employ the Adam optimizer to update $U_A$ iteratively until convergence for

each of the 100 samples of $V_1, V_2$. We consider the converged value as a good estimation with a tolerable error at least for circuits with a small number of qubits ($\leq 10$) and a modest depth ($\leq 10 \times n$). We repeat this procedure for different numbers of qubits and different statistics of $\mathbb{V}_1$ and $\mathbb{V}_2$, i.e., $\mathbb{V}_1$ or $\mathbb{V}_2$ being a 2-design (1-design) while the other being identity.

**Numerical Results.** We summarize the simulation results of the three examples in Fig. 5. The slopes of the lines imply the rates of exponential decay. The data points represent the sample averages of the cost variation range $\Delta_{H,\rho}$ via varying $U_A$ over $w(H)$, and the error bars represent the standard deviations over samples. We specially rescale the error bar in the QSL example as a quarter of the standard deviation for better presentation on semi-log plots. One can see that in all the cases, the expectations of $\Delta_{H,\rho}(V_1, V_2)$ vanish exponentially in the number of qubits. The data lines are almost parallel to the dashed lines depicting the theoretical upper bounds. That is to say, the scaling behaviors almost coincide with the predictions from Theorem 1 and Proposition 2. These results suggest that while optimizing a local unitary within a random QNN, the cost function exhibits fluctuations within an exponentially small range relative to the number of qubits. It is this phenomenon that elucidates the vanishing gradient issue and contributes to the exponential difficulty of training as the QNN scales up. A detailed derivation can be found in Appendix B for the tighter task-dependent upper bounds used in Fig. 5(a) and (b).

## 7   Conclusion and Discussion

We have shown that the maximal possible influence of a local unitary within a QNN on the cost function vanishes exponentially in the number of qubits with a high probability. This finding unveils the exponential hardness associated with training QNNs as they scale up. The randomness required is just a 2-design for the generic cost function and a 1-design for the fidelity-type cost function, in spite that the integrand $\Delta_{H,\rho}(V_1, V_2)$ is not necessarily a polynomial of degree at most 2 or 1 in the entries of $V_1$ and $V_2$. We remark that a 2-design circuit can be achieved approximately by only $\mathcal{O}(n)$ depth [25, 62, 63] for common repeated-layer-type ansatzes, e.g., the hardware-efficient ansatzes [61], and a 1-design circuit can be achieved more easily by only $\mathcal{O}(1)$ depth.

From the perspective of quantum information theory, our results can be regarded as a basic property of random quantum circuits. That is, a local unitary within a random circuit of polynomial depth has an exponentially small impact on the expectation of physical observables, which is expected to have potential applications in other areas involving random quantum circuits. This property may also provide insight into QNN design to address the critical trainability issue.

For the training of QNN, our results unify the restrictions on gradient-based and gradient-free optimizations in a natural way and hence can be regarded as the underlying mechanism behind the barren plateau phenomenon. Therefore, a fundamental limitation is unraveled in training QNNs, which can serve as a guide for designing better training strategies to improve the scalability of QNNs. A direct consequence is that the gate-by-gate optimization strategy [39, 40] is ineffective no matter what optimizers are utilized. Reparameterization within local unitaries is also unhelpful. For future research, it will be of great interest to explore potential solutions via proper initialization [41], pre-training including adaptive methods [42–46], circuit architectures [47, 48] and cost function choices [49, 50].

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

541 05820.

