# OpenReview forum: "Exponential Hardness of Optimization from the Locality in Quantum Neural Networks"
_NeurIPS.cc/2023/Conference — Submitted to NeurIPS 2023_

### Official Review · Reviewer_dzQA · 2023-06-30

**Soundness:** 3 good
**Presentation:** 3 good
**Contribution:** 3 good
**Rating:** 7
**Confidence:** 4

**Summary:**

In this work, the authors characterize the problem of the Barren Plateau from different perspectives: (1) local unitary within a QNN on the cost function, particularly the randomness for the generic cost function; (2) quantum information theory; (3) the optimization methods during training. This work discusses those factors impacting the Barren Plateau landscape.

**Strengths:**

(1) The work provides a theoretical understanding of the Barren Plateau problem and determines what factors actually impact the training of VQC, which is very interesting.

(2) Solid mathematical formulation is given and the experiments can corroborate the theoretical analysis.


**Weaknesses:**

(1) Some latest work on the Barren Plateau problem in the training of VQC should be included, such as Refs. [1], [2], and [3]. Ref. [1] aims at the QNN architecture for dealing with the Barren Plateau problem, Ref. [2] focuses on the initialization strategy, and Ref. [3] puts forth the pre-training method for mitigating the VQC training problem of Barren Plateau.

[1] Jun Qi, Chao-Han Huck Yang, Pin-Yu Chen, Min-Hsiu Hsieh, "Theoretical Error Performance Analysis for Variational Quantum Circuit Based Functional Regression," npj Quantum Information, Vol. 9, no. 4, 2023

[2] Zhang, Kaining, Hsieh, Min-Hsiu, Liu, Liu, and Tao, Dacheng. Gaussian Initializations Help Deep Variational Quantum Circuits Escape From the Barren Plateau. In Neural Information Processing Systems, 2022.

[3] Jun Qi, Chao-Han Huck Yang, Pin-Yu Chen, Min-Hsiu Hsieh, "Pre-Training Tensor-Train Networks Facilitate Machine Learning with Variational Quantum Circuits," arXiv:2306.03741v1

**Questions:**

How to asscoated with the locality unitary property with other methods for dealing with Barren Plateau problem, such as the DNN architecture and pre-training method?

**Limitations:**

It is expected to have experimental simulations on real data like the MNIST dataset to demonstrate the effectiveness of the proposed analysis approach.

---

> ### Author Rebuttal · Authors · 2023-08-09
>
> We greatly thank the reviewer for their time and the helpful feedback. Here we respond to the comments and questions.
>
> > $\textbf{Comment 1:}$ ``Some latest work on the Barren Plateau problem in the training of VQC should be included, such as Refs. [1], [2], and [3]. Ref. [1] aims at the QNN architecture for dealing with the Barren Plateau problem, Ref. [2] focuses on the initialization strategy, and Ref. [3] puts forth the pre-training method for mitigating the VQC training problem of Barren Plateau.''
>
> $\textbf{Re 1:}$ Great thanks for bringing these recent works on the QNN training problem to our attention. We will ensure to include and discuss these references to provide more comprehensive research in our revision.
>
> ---
> > $\textbf{Comment 2:}$ ``How to associated with the locality unitary property with other methods for dealing with Barren Plateau problem, such as the DNN architecture and pre-training method?''
>
> $\textbf{Re 2:}$ We appreciate the reviewer's thoughtful comment. Our theorem is mainly based on two assumptions. (a) The QNN is composed of local quantum gates. (b) The QNN is randomly initialized and approximate a 2-design. Thus to enhance the trainability, one can construct global parameterized gates by e.g. correlating parameters to break assumption (a). On the other hand, one can design problem-tailored QNN architecture or use pre-training to escape from being a 2-design to break assumption (b). We will address this important aspect in our revision.

---

> > ### Comment · Reviewer_dzQA · 2023-08-17
> > **Follow-up for the rebuttal letter**
> >
> > Thank the authors for the rebuttal letter. The authors' responses have perfectly resolved my major concerns. Since the trainability of QNN is a significant issue, this paper has a significant contribution in this aspect. So, I highly recommend this paper be accepted by NeurIPS.

---

### Official Review · Reviewer_n6Yp · 2023-07-02

**Soundness:** 3 good
**Presentation:** 3 good
**Contribution:** 2 fair
**Rating:** 4
**Confidence:** 5

**Summary:**

The paper examines the critical issue of trainability in quantum neural networks (QNNs) by adopting a perspective centered around the locality. Through extensive analysis, the authors convincingly demonstrate that the adjustment of local quantum gates within a diverse range of QNNs results in an exponential decay of the loss function range as the number of qubits scales up. The authors bolster their claims with carefully conducted numerical simulations, providing compelling evidence that locality plays a fundamental role in shaping the behavior of QNNs. Building upon prior research on barren plateaus, the paper makes a technically sound contribution, albeit with an incremental advancement in the field.


**Strengths:**

The analysis of Theorems and Propositions, which shows the exponential decay of the loss function range by adjusting local quantum gates, is technically sound. Additionally, the ideas, concepts, and results are well presented. The authors effectively communicate their methodology, theoretical framework, and experimental simulations, making it easier for readers to comprehend and follow their arguments.









**Weaknesses:**

The main weakness of this paper lies in its limited impact. While the authors conduct a clear and thorough analysis of how the concentration results of random circuits depend on the locality unitary, the technical tools employed bear a striking resemblance to prior literature concerning barren plateaus. The achieved results can be derived from existing works, with the only notable distinction being the introduction of a parameter, m, related to the locality in the derived bound. Additionally, the authors' claim that few rigorous scaling results exist for generic QNNs is contradicted by the abundance of relevant research, as evidenced by references [1], [2], [3], and [4], which address similar theoretical aspects the authors aim to explore. Previous studies have already established that deep ansatz can lead to the concentration of the cost function, rendering the observation regarding the exponential vanishing of the loss function range via the adjustment of local quantum gates less novel.


[1] Leone, Lorenzo, et al. "On the practical usefulness of the Hardware Efficient Ansatz." arXiv preprint arXiv:2211.01477 (2022).
[2] Thanasilp, Supanut, et al. "Subtleties in the trainability of quantum machine learning models." Quantum Machine Intelligence 5.1 (2023): 21.
[3] Garcia, Roy J., et al. "Barren plateaus from learning scramblers with local cost functions." Journal of High Energy Physics 2023.1 (2023): 1-79.
[4] Larocca, Martin, et al. "Diagnosing barren plateaus with tools from quantum optimal control." Quantum 6 (2022): 824.

**Questions:**

- The results presented in Theorem 4 are somewhat perplexing. It appears that the upper bound can be independent of the qubit number n when utilizing the global unitary U_A. For example, when m=n, the concentration phenomenon can be avoided. I am not sure if I have overlooked certain details.

- In Remark 2, what is the necessity of decomposing and preserving the local unitary for subsequent analysis of U_A? A more detailed explanation is expected.

- The similar findings discussed in Section 4 can be derived from existing research. Specifically, the concentration phenomenon of the cost function, as outlined in reference [5], in conjunction with the parameter shift rule (for first-order and higher-order) or the finite-difference method (for zero-order), can effectively yield similar outcomes as presented in Section 4. It would be valuable if the authors could acknowledge and discuss these connections with prior findings to provide a clearer context and further highlight the novelty of their results. If the only difference lies in the introduced parameter related to the locality m?

[5] Arrasmith, Andrew, et al. "Equivalence of quantum barren plateaus to cost concentration and narrow gorges." Quantum Science and Technology 7.4 (2022): 045015.

**Limitations:**

No, the authors did not address the limitations of their work

---

> ### Author Rebuttal · Authors · 2023-08-09
>
> We thank the reviewer for the helpful feedback and for acknowledging that our paper is technically sound and well-presented. Below is a detailed response to the questions raised by the reviewer.
>
> > $\textbf{Comment 1:}$ ``The main weakness of this paper lies in its limited impact ...''
>
> $\textbf{Re 1:}$ Thank you for your feedback and we will make the following clarifications and enhancements to the paper:
>
> i): Prior Literature:
> We thank the reviewer for pointing out other relevant research concerning the scaling results of QNNs. We will discuss the prior literature (Ref. [1], [2], [3], and [4]) in the final version. We would like to clarify our differences with prior literature. Most of the previous results for characterizing the QNN training are related to the vanishing gradient. Whereas we present the maximum variation range of the cost function when optimizing a local unitary. Our theorem $\textbf{can not be derived}$ from existing works. For instance, the process of taking extreme values of the cost function in our work is not involved in other works considering the cost function difference.
>
> ii): Novelty and Impact: the study of QNN training is pivotal for harnessing the potential of quantum computing. Different from the celebrated barren plateau phenomena, our paper delivers a new understanding of the training landscape from a more intuitive way. We agree with the reviewer that previous studies have already established the concentration of the cost function, which are mainly from the perspective of the gradient of the QNN parameters. However, we believe that the entire variation range of the cost function when you are optimizing a local gate has an intuitive meaning in QNN training. Its property can not be derived from the existing works. Besides the implications in VQAs, our results can be regarded as a basic property of random quantum circuits.
>
> [1] Leone, Lorenzo, et al. "On the practical usefulness of the Hardware Efficient Ansatz." arXiv preprint arXiv:2211.01477 (2022).
>
> [2] Thanasilp, Supanut, et al. "Subtleties in the trainability of quantum machine learning models." Quantum Machine Intelligence 5.1 (2023): 21.
>
> [3] Garcia, Roy J., et al. "Barren plateaus from learning scramblers with local cost functions." Journal of High Energy Physics 2023.1 (2023): 1-79.
>
> [4] Larocca, Martin, et al. "Diagnosing barren plateaus with tools from quantum optimal control." Quantum 6 (2022): 824.
>
> ---
> > $\textbf{Comment 2:}$ ``The results presented in Theorem 4 are somewhat perplexing ...''
>
> $\textbf{Re 2:}$ Thanks for this careful question. We apologize for any confusion resulting in the reference to ``Theorem 4''. In case you are asking the Theorem 1, we would like to explain it as follows. We agree that the utilization of global unitary $U_A$ can lead to a trivial upper bound independent of the qubit number $n$. This is because we define the variation range by taking optimum over all possible unitaries $U_A$ with support $m$. Thus it is not surprising that a universal global unitary can lead to a non-vanishing range.  In this case, our result indeed does not indicate the limitation of training, but parameterizing and optimizing such a universal unitary is impractical. However, the unitaries we can optimize are usually single-qubit and two-qubit quantum gates in a practical cases, corresponding to the case $m=1,2$, where our theorem gives an exponentially small upper bound.
>
> ---
> > $\textbf{Comment 3:}$ ``In Remark 2, what is the necessity of decomposing and preserving the local unitary for subsequent analysis of U_A? ...''
>
> $\textbf{Re 3:}$ Thanks for this careful question. Remark 2 points out that the exponential small bound in Theorem 1 with $m=1$ can be extended to the case where $U_A$ is a global unitary satisfying the parameter-shift rule. However, for the case where $U_A$ does not satisfy the parameter-shift rule, such as the controlled Pauli rotation gates, Remark 2 does not generally hold and we have only Theorem 1. We will make this point more clear in our revised manuscript.
>
> ---
> > $\textbf{Comment 4:}$ ``The similar findings discussed in Section 4 can be derived from existing research ...''
>
> $\textbf{Re 4:}$ Thank you for the careful comment regarding the similarity of our findings in Section 4 to existing research and the potential connections with prior work, particularly Ref. [5]. We appreciate the opportunity to address this concern and clarify the context of our results. Ref. [5] considers the cost function difference between two points either both randomly chosen or one random and the other has a deterministically chosen distance with it. Whereas we consider the difference between the maximum and minimum within the whole subspace w.r.t. a local unitary. The process of taking extreme values in our work is not involved there. Their relation is further clarified in Eq.(9) after line 198 in our paper. Actually, we also acknowledge the work of Ref.[6] which presents relevant results as you may concern.
>
> [6] Andrew Arrasmith, M. Cerezo, Piotr Czarnik, Lukasz Cincio, and Patrick J. Coles, “Effect of barren plateaus on gradient-free optimization,” Quantum 5, 1–9 (2020).

---

> > ### Comment · Reviewer_n6Yp · 2023-08-19
> >
> > Thank you for your response. After carefully reviewing the authors' feedback, I have some additional comments to share.
> >
> > Paper's Impact: Let me showcase how to use existing works to attain a similar result achieved in the paper. For simplicity, denote the cost function in Definition 1 as $C(\theta)$, where $U_A = \exp(-i\theta H)$. Then, applying Taylor expansion to the cost function, we obtain $$C(\theta)=C(\theta_0)+C'(\theta_0)(\theta-\theta_0)+\frac{C''(\theta_0)}{2!}(\theta-\theta_0)^2+\frac{C^{(3)}(\theta_0)}{3!}(\theta-\theta_0)^3+...+\frac{C^{(n)}(\theta_0)}{n!}(\theta-\theta_0)^n.$$ This expansion allows us to establish a connection between the cost function range examined in this paper and previous literature. Recall that the gradient or higher-order derivatives of the cost function tend to exponentially approach zero with the increase in the number of qubits [M. Cerezo and Patrick J. Coles, Quantum Science and Technology 6, 035006 (2021)], it becomes conceivable that the variability range of the cost function also experiences an exponential diminishment as the qubit count grows, i.e., $|C(\theta)-C(\theta')|\rightarrow O(\exp(-n))$.
> >
> > I acknowledge the technical endeavors undertaken by the authors, which might not have been directly deduced from existing literature. However, it's worth noting that there may exist multiple straightforward avenues to achieve similar insight (at least in a rough sense) through the utilization of results obtained from prior works.
> >
> > Reply 2 Clarification: In Reply 2, the authors assert that *the unitaries suitable for optimization typically pertain to single-qubit and two-qubit quantum gates in practical scenarios, corresponding to the cases of m=1 and m=2. Here, our theorem offers an exponentially small upper bound*. I respectfully hold a divergent perspective on this matter. A wide array of ansatzes, such as the quantum approximate optimization ansatzes and group-invariant ansatzes, typically align with the scenario where $m\gg 2$. From this perspective, Considering this angle, it becomes important to engage in a thorough discussion regarding the broader impact and applicability of the obtained results.

---

> > > ### Author Response · Authors · 2023-08-19
> > >
> > > We greatly appretiate the reviewer's thoughtful comments. After carefully reviewing these additional comments, we provide our response as follows.
> > >
> > > > $\textbf{Comment~1:}$ ``Paper's Impact: Let me showcase how to use existing works ...''
> > >
> > > $\textbf{Re 1:}$
> > > The reviewer provides a rough argument using Taylor expansion in order to show that the cost function difference $C(\theta)-C(\theta')$ between two parameter points is exponentially small given exponentially small derivatives. In fact, this idea has already been formulated in a more strict manner by Ref. [26] mentioned in our manuscript (line 175~208). However, we emphasize that the cost function difference
> > > $ C(\theta)-C(\theta'),$ is very different from the variation range
> > > $$
> > > \max_{U_A} C(\mathbf{U})-\min_{U_A} C(\mathbf{U}),
> > > $$
> > > studied in our work. Because the former focuses on two *fixed* parameter points (independent with the probability ensemble) while the latter takes the possible maximal range over all parameters within $U_A$. In other words, the latter implies that the cost function difference between any two parameter points *simultaneously* vanishes, which is beyond the former result.
> > >
> > > Furthermore, we would like to emphasize that the significance of our work lies not only in providing technical outcomes, but also in its foundational implication. Our work unifies the restrictions on gradient-based and gradient-free optimizations from a new perspective which is independent with gate parameterization, reveals a fundamental property of random quantum circuits and deepens the understanding of the role of locality in QNNs.
> > >
> > > [26] Andrew Arrasmith, M. Cerezo, Piotr Czarnik, Lukasz Cincio, and Patrick J. Coles. Effect of barren plateaus on gradient-free optimization. Quantum, 5:1–9, nov 2020. ISSN 2521327X. doi:10.22331/q-2021-10-05-558.
> > >
> > > > $\textbf{Comment~2:}$ ``Reply 2 Clarification: In Reply 2, the authors assert ...''
> > >
> > > $\textbf{Re~2:}$
> > > We appreciate your perspective and understand the importance of encompassing a wide range of ansatzes and scenarios in the discussion of our results' applicability. We would like to offer some further clarifications to address your concerns.
> > >
> > > Firstly, it's important to underscore that our theorem involves a scaling relationship of the gate locality $m$ with respect to the qubit count $n$. And our key emphasis lies in the situation of local gate where $m$ does not scale with $n$ so that our theorem can give a non-trivial exponential upper bound. This is indeed a situation frequently encountered in practice like in variational quantum eigensolver and quantum state learning, considering the fact that common elementary quantum gates available on digital quantum computers (e.g., Pauli rotation gates and CNOT gates) are inherently local. This aligns with the hardware constraints and technological advancements shaping current quantum devices.
> > >
> > > Secondly, we agree with the reviewer that there are indeed instances utilizing global unitaries, like QAOA, which is beyond the scope of our theorem. These ansatzes usually realizes a parameterized global unitary by correlating parameters among many local parameterized gates. This is actually one of the implications of our theorem. That is to say, one possible strategy to escape from the vanishing variation range of the cost function is correlating parameters in multiple local gates like in QAOA. We value the reviewer's insights in this regard and will certainly engage in a more comprehensive discourse concerning the broader impact and applicability of our derived results in the upcoming revised version of our paper.

---

### Official Review · Reviewer_T1RK · 2023-07-04

**Soundness:** 3 good
**Presentation:** 3 good
**Contribution:** 2 fair
**Rating:** 5
**Confidence:** 3

**Summary:**

This paper investigates the trainability of random quantum circuits from the perspective of their locality and demonstrates the variation range of the cost function via adjusting any local quantum gate vanishes exponentially in the number of qubits. This theorem unifies the restrictions on gradient-based and gradient-free optimizations. The paper also verifies their theorem on three applications with numerical simulations and deepens the understanding of the role of locality in QNNs.

**Strengths:**

1. The paper is well-written and provides a rigorous analysis of QNN trainability and scalability from the perspective of their locality.
2. The paper applies the proposed theorem to three representative QNN models, including the VQE, quantum autoencoder, and quantum state learning, and provides the numerical simulation results.

**Weaknesses:**

1.  Although Line 66-73 provides the advances of the proposed method, the comparison with previous works is not clear enough. It is important to review the previous methods and compare their specific differences.
2.  The contribution of the paper seems weak, the novelty, comparisons with related works, and guidances for future QNN training or design need to be highlighted and enhanced.

**Questions:**

Compared with previous approaches that prove barren plateaus, what new enlightenment does the proposed method bring to the design and training of QNN? Specifically, barren plateaus are a well-known phenomenon in the field of quantum machine learning, and we usually design QNNs with single-qubit and tow-qubit parametric gates for practicality on NISQ devices, and it seems to have defaulted to the property of barren plateaus w.r.t local unitary. What guidance can the proposed finding give to the design of QNNs? Moreover, based on the finding, what are some specific suggestions to design more effective training strategies?

**Limitations:**

The paper can be improved by considering the above weaknesses and questions.

---

> ### Author Rebuttal · Authors · 2023-08-09
>
> We greatly appreciate the reviewer's recognition of our work as technically solid and well written. We also thank the reviewer for the helpful feedback. A detailed response to the reviewer's comments and questions is provided below.
>
> > $\textbf{Comment 1:}$ ``Although Line 66-73 provides the advances of the proposed method, the comparison with previous works is not clear enough. It is important to review the previous methods and compare their specific differences.''
>
> $\textbf{Re 1:}$ Thanks for your comments regarding the unclearness of the comparison between our work and previous work. The specific differences between our work and previous work are as follows.
>
> i). Ref. [1],[2],[3] analyze the gradient of parameters in QNNs. Basically, the results for characterizing the QNN training are related to the vanishing gradients along any reasonable direction.
> Whereas our work does not start with calculating the gradients. We present the maximum variation range of the cost function when you optimize a local unitary that has not been analyzed before. We believe this is practically valuable regarding the optimization strategy and also theoretically meaningful in studying random quantum circuits.
>
> ii) Ref. [4] also considers the cost function difference. However, the quantity they consider is the difference between two points either both randomly chosen or one random and the other has a deterministically chosen distance with it. Again, we consider the difference between the maximum and minimum within the whole subspace w.r.t. a local unitary. The process of taking extreme values of the cost function in our work is not involved in Ref. [4].
>
> [1]. Jarrod R. McClean, Sergio Boixo, Vadim N. Smelyanskiy, Ryan Babbush, and Hartmut Neven, “Barren plateaus in quantum neural network training landscapes,” Nature Communications 9, 1–7 (2018).
>
> [2] Garcia, Roy J., et al. "Barren plateaus from learning scramblers with local cost functions." Journal of High Energy Physics 2023.1 (2023): 1-79.
>
> [3] M. Cerezo and Patrick J. Coles, Quantum Science and Technology 6, 035006 (2021).
>
> [4] Arrasmith, Andrew, et al. Quantum Science and Technology 7.4 (2022): 045015.
>
> ---
> > $\textbf{Comment 2:}$ ``The contribution of the paper seems weak, the novelty, comparisons with related works, and guidances for future QNN training or design need to be highlighted and enhanced.''
>
> $\textbf{Re 2:}$ We greatly appreciate the reviewer's feedback on the contribution of our paper. In what follows, we would like to explain the novelty, comparisons with related works, and guidances for QNN training or designing.
>
> i): For the novelty and comparisons with related works, we clarify that the main quantity we focus on in this work is the entire variation range of the cost function via adjusting any local unitary within the circuit, which has not been analyzed before. And our results unify the restrictions on gradient-based and gradient-free optimizations. Such a quantity has an intuitive meaning in QNN training whose properties can not be derived from the existing works.
>
> ii): One direct guidance for QNN training and designing is that the gate-by-gate optimization strategy (which tries to avoid barren plateaus using gradient-free optimization for each gate) is ineffective no matter what optimizers are utilized. Reparameterization within local unitaries is also unhelpful. Our theorem is mainly based on two assumptions. (a) The QNN is composed of local quantum gates. (b) The QNN is randomly initialized and approximate a 2-design. Thus to enhance the trainability, one can construct global parameterized gates by e.g. correlating parameters to break assumption (a). On the other hand, one can design problem-tailored QNN architecture or use pre-training to escape from being a 2-design to break assumption (b). We will address this important aspect in our revision.
>
> ---
> > $\textbf{Comment 3:}$ ``Compared with previous approaches that prove barren plateaus, what new enlightenment does the proposed method bring to the design and training of QNN? ...''
>
> $\textbf{Re 3:}$ Thanks for this very good question. Indeed, barren plateaus is a well-known phenomenon in the field of QML which limits the training of the QNNs. However, phenomenological calculations such as gradient analyses and their descendent have not unraveled enough information on the training landscape. The quantity we consider is the entire variation range of the cost function when you are optimizing a local gate. We believe this quantity has an intuitive meaning in QNN training whose properties can not be derived from the existing works. Also, its importance lies in the unification of the existing explanations of the hardness of QNN training.
>
> Our results offer new enlightenment in two aspects, both theoretical and practical. Theoretically, we show that the effect of a local operation on a physical observable will vanish exponentially after a chaotic evolution. In practice, a direct consequence is that the gate-by-gate optimization strategy (which tries to avoid barren plateaus using gradient-free optimization for each gate) is ineffective no matter what optimizers are utilized. Reparameterization within local unitaries is also unhelpful. Our theorem primarily relies on two underlying assumptions. Firstly, the QNN is built using local quantum gates. Secondly, the QNN is initialized randomly and approximates a 2-design. Therefore, for the purpose of improving trainability, one approach is creating global parameterized gates by, for instance, establishing correlations among parameters. Conversely, to deviate from being a 2-design, one could devise a QNN architecture tailored to a specific problem or employ pre-training strategies. We will make this point more clear in the revision.

---

> > ### Comment · Reviewer_T1RK · 2023-08-20
> >
> > Thanks for your answers. I have no further questions and believe the quality of the paper will improve if the above discussion can be incorporated into the revised paper. I have adjusted my score to reflect this (4 -> 5).

---

### Official Review · Reviewer_J4bd · 2023-07-04

**Soundness:** 3 good
**Presentation:** 3 good
**Contribution:** 2 fair
**Rating:** 6
**Confidence:** 3

**Summary:**

The paper proof a result on the range of possible values that the cost function of a variational quantum algorithm can take when one optimises over a given unitary that is before or after random gates that form unitary 2 designs. This quantity vanishes exponentially with the number of qubits. This generalises previous results on Barren plateaus, concerned with the vanishing of the gradient of the cost function.
The material is presented clearly and the paper also has numerical verification of the scaling in the case of VQE, quantum autoencoder and quantum state learning.

**Strengths:**

- The paper is clearly written, with figures explaining concepts. It presents both theory and numerical checks
- The problem studied is relevant in scaling up quantum neural networks
- The main theorem allows the authors to recover and unify previous results on exponentially vanishing gradients and cost function differences

**Weaknesses:**

- The paper does not comment on recommendations to avoid the exponentially vanishing variation range
- The comparison to previous works is limited. The authors mention that their work opens a new venue for analysing trainability of QNNs but it is not clear to me what new insights are gained. It would be useful to comment on what is gained wrt previous literature. Also, on whether the methods used to prove their main theorem are similar to those used in the literature or not.
- The VQE experiments are taken with circuits of depth 10 x n. That depth was chosen so that the hardware aware ansatz approximates a 2-design. However no comment on the required depth to compute the ground state of the Hamiltonian is presented and it is not clear whether the choice of ansatz and depth is something that practitioner would actually do.

Minor

- Sentence Line 71 - 73 does not read very well, you could rephrase it
- Line 153 - 155: it would be helpful for the reader to have an explanation of the connection between parameter shift rule and $e^{-i\theta \Omega}$ with $\Omega^2=1$. Also why does this imply the existence of $W$ as claimed?

**Questions:**

- Can you comment on your choice of VQE ansatz explaining whether 10x n is needed or smaller depth would be enough to solve this problem? Also, do other ansatz, such as alternating ansatz, suffer from exponentially vanishing variation range?
- What are the additional insights that you get on top of previous works for quantum neural network trainability?
- Can you expand on the previous literature? Do the previous results also require approximate 2-design?

**Limitations:**

- The method applies when the $V_1$ or $V_2$ are 2-designs. This limits applicability.
- Strategies to overcome the exponentially vanishing range are not discussed.

---

> ### Author Rebuttal · Authors · 2023-08-09
>
> We greatly appreciate the reviewer's positive assessment on the correctness and significance of our work. We also thank the reviewer for the very helpful feedback. Below is our point-by-point response to the comments and questions.
>
> > $\textbf{Comment 1:}$ ``The paper does not comment on recommendations to avoid the exponentially vanishing variation range.''
>
> $\textbf{Re 1:}$ Thanks for the valuable feedback. Problem-inspired architectures are highly recommended since prior information is used to escape from being a 2-design. We will include more related discussion in our revision.
>
> ---
>
> > $\textbf{Comment 2:}$ ``The comparison to previous works is limited. The authors mention that their work opens a new venue for analysing trainability of QNNs but it is not clear to me what new insights are gained. It would be useful to comment on what is gained wrt previous literature. Also, on whether the methods used to prove their main theorem are similar to those used in the literature or not.''
>
> $\textbf{Re 2:}$ The main differences are summarized as follows.
>
> i): Most previous works focus on the gradient which usually provides limited information around the vicinity, while our work analyzes certain entire variation ranges when tuning local unitary. This brings new insight that the locality of QNN also plays an import role in the trainability limitation.
>
> ii): Besides the standard Haar integration, the proof involves new techniques including the Hamiltonian splitting, diverse norm inequalities in quantum information theory, as well as tensor network diagrams to compute high-order integrals.
>
> ---
> > $\textbf{Comment 3:}$ ``The VQE experiments are taken with circuits of depth 10 x n. That depth was chosen so that the hardware aware ansatz approximates a 2-design. However no comment on the required depth to compute the ground state of the Hamiltonian is presented and it is not clear whether the choice of ansatz and depth is something that practitioner would actually do.''
>
> $\textbf{Re 3:}$ Thanks for highly practical comment. $10\times n$ is a empirically chosen depth which is also discussed in previous works [1-3]. Unfortunately, the practically required depth in VQE depends on the specific Hamiltonian and there is a lack of universal guarantee to estimate the depth tailored for each Hamiltonian. We will highlight the discussion of this part in the revision.
>
> [1] Jarrod R. McClean, et al. Nature Communications 9, 1–7 (2018).
>
> [2] Aram W. Harrow, et al. Physical Review Letters 103, 150502 (2009).
>
> [3] Fernando G. S. L. Brandão, et al. Communications in Mathematical Physics 346, 397–434 (2016).
>
> ---
> > $\textbf{Comment 4:}$ ``Sentence Line 71 - 73 does not read very well, you could rephrase it.''
>
> $\textbf{Re 4:}$ Many thanks for your careful comment. We will improve the statement in our revision.
>
> ---
> > $\textbf{Comment 5:}$ ``Line 153 - 155: It would be helpful for the reader to have an explanation of the connection between parameter shift rule and $e^{-i\theta\Omega}$ with $\Omega^2=I$. Also why does this imply the existence of $W$ as claimed?''
>
> $\textbf{Re 5:}$ Thanks a lot for this careful question. The condition $\Omega^2=I$ ensures the equality $e^{-i\theta\Omega}=I\cos\theta - i\Omega\sin\theta$ and hence the expectation value w.r.t a single parameter must be some triangular function, the derivative of which can be exactly expressed as a finite difference, i.e., satisfying the parameter-shift rule. The conditions $\operatorname{tr}(\Omega)=0$ and $\Omega^2=I$ imply that half of the eigenvalues of $\Omega$ is $1$ and the other half is $-1$. Thus $\Omega$ has the same eigen spectrum with, e.g., the local operator $Z\otimes I\otimes \cdots\otimes I$, so that there must exist a unitary $W$ to diagonalize $\Omega$. We will make this point more clear in the revision.
>
> ---
> > $\textbf{Comment 6:}$ ``Can you comment on your choice of VQE ansatz explaining whether 10x n is needed or smaller depth would be enough to solve this problem? Also, do other ansatz, such as alternating ansatz, suffer from exponentially vanishing variation range?''
>
> $\textbf{Re 6:}$ The proximity between a randomly initialized ansatz and a 2-design can serve as a measure of its expressibility [4]. $10\times n$ is an empirically chosen depth [1-3]. Regarding the other ansatzes, all the ansatzes that form 2-design will suffer from the exponentially vanishing variation range. Some problem-inspired ansatzes with specially designed architectures may avoid the vanishing variation range problem because of escaping from the zone of 2-design using prior information.
>
> [4] Holmes, Zoë, et al. PRX Quantum 3.1 (2022): 010313.
>
> ---
> > $\textbf{Comment 7:}$ ``What are the additional insights that you get on top of previous works for quantum neural network trainability?''
>
> $\textbf{Re 7:}$ Most of the previous works are based on gradient analysis, which require specific parameterization like $e^{-i\Omega\theta}$. However, here we focus on the vanishing variation range of the cost function. Our result is a fundamental property of random quantum circuits regardless of parameterization, which rules out all optimization methods focusing on optimizing local quantum gates independently.
>
> ---
> > $\textbf{Comment 8:}$ ``Can you expand on the previous literature? Do the previous results also require approximate 2-design?''
>
> $\textbf{Re 8:}$ Thanks for your question. The requirement of 2-design is a common assumption in analyzing QNNs [5-7]. Barren plateaus was first presented using 2-design random circuits. The expressibility was also shown to have a close connection with its distance to 2-design [4]. We will make this point more clear in the revision.
>
> [5] Marrero, Carlos Ortiz, Mária Kieferová, and Nathan Wiebe. PRX Quantum 2.4 (2021): 040316.
>
> [6] Sack, Stefan H., et al. PRX Quantum 3.2 (2022): 020365.
>
> [7] Holmes, Zoë, et al. Physical Review Letters 126.19 (2021): 190501.

---

> > ### Comment · Reviewer_J4bd · 2023-08-17
> >
> > Thank you for the rebuttal and clarifications. I do not have further comments.

---

### Author Rebuttal · Authors · 2023-08-10

Dear PC,

We are grateful to the PCs for their efforts in shaping the conference's scientific program and to the reviewers for their dedicated time and efforts in reviewing our paper.

We thank reviewers J4bd and dzQA for recognizing our paper's sound technique and clear writing style, acknowledging its potential to contribute to the field, and both recommending acceptance.

Reviewer T1RK is concerned about novelty and significance, questioning the comparison with related works and our contribution to the existing body of knowledge. In the rebuttal, we clarified that our paper considers a novel feature of the QNN training landscape, and one key distinction is that our results unify the restrictions on gradient-based and gradient-free optimizations in training variational quantum circuits. We also explained our differences with previous results precisely and highlighted what new enlightenment our results deliver to the study of QNN.

Reviewer n6Yp has concerns with our paper's impact and comparison with previous results on QNN training limitations. In the rebuttal, we explained the main differences between our results and prior research. Our main result on the entire variation range of the cost function has an intuitive meaning in QNN training, and such property can not be derived from the existing works. As recognized by reviewers J4bd and dzQA, our work established a theoretical understanding of the Barren Plateau problem and determined the behaviors of training QNN via gradient-based and gradient-free methods, which is significant to the study of quantum machine learning. Besides the implications in QNN training, our results can be regarded as a basic property of random quantum circuits, which is expected to have potential applications in other areas involving random quantum circuits.

We have addressed the reviewers' comments in the rebuttal. We want to thank all the PCs and ACs for their time and efforts, regardless of the result of this paper.

Yours Sincerely,

Authors of Paper 10546.

---

### Decision · Program_Chairs · 2023-09-21

**Decision:**

Reject

**Comment:**

The paper examines the hardness of trainability in quantum neural networks (QNNs) due to the locality. Through extensive analysis, the authors demonstrate that the adjustment of local quantum gates within a diverse range of QNNs results in an exponential decay of the loss function range as the number of qubits scales up.  One major concern is about the technical novelty of the result. In particular, the authors are advised to examine whether a simple argument based on the existing literature could lead to a similar result, in light of the communication that happened during the rebuttal phase.  In general, the paper should identify and articulate the precise relation of their results to the prior art.  Moreover,  the paper could also benefit from a more comprehensive discussion of the implications of this hardness result.